# Neurotrophins Time Point Intervention after Traumatic Brain Injury: From Zebrafish to Human

**DOI:** 10.3390/ijms22041585

**Published:** 2021-02-04

**Authors:** Pietro Cacialli

**Affiliations:** Department of Pathology and Immunology, University of Geneva, Rue Michel-Servet 1, 1211 Geneva 4, Switzerland; pietro.cacialli@unige.ch

**Keywords:** zebrafish, brain, injury, BDNF, NGF, NT3, NT4, TrkA, TrkB, TrkC

## Abstract

Traumatic brain injury (TBI) remains the leading cause of long-term disability, which annually involves millions of individuals. Several studies on mammals reported that neurotrophins could play a significant role in both protection and recovery of function following neurodegenerative diseases such as stroke and TBI. This protective role of neurotrophins after an event of TBI has also been reported in the zebrafish model. Nevertheless, reparative mechanisms in mammalian brain are limited, and newly formed neurons do not survive for a long time. In contrast, the brain of adult fish has high regenerative properties after brain injury. The evident differences in regenerative properties between mammalian and fish brain have been ascribed to remarkable different adult neurogenesis processes. However, it is not clear if the specific role and time point contribution of each neurotrophin and receptor after TBI is conserved during vertebrate evolution. Therefore, in this review, I reported the specific role and time point of intervention for each neurotrophic factor and receptor after an event of TBI in zebrafish and mammals.

## 1. Traumatic Brain Injury Incidence

Traumatic brain injury (TBI), according to the World Health Organization (WHO), could surpass many diseases as the major cause of death and disability over the next years. TBI is one of the leading causes of disability in Europe and the United States, estimated at 13–14 million individuals [1,2]. Many survivors live with significant disabilities, resulting in major socioeconomic burden as well [3]. Previous report showed that Latin America and Sub-Saharan Africa demonstrate a higher TBI-related incidence rate, namely, 150 and 170 per 100,000, respectively, due to rate of traumatic incidence RTIs compared to a global rate of 106 per 100,000. Further studies also showed that there is a large gap in the data on incidence, risk factors, sequelae, financial costs, and social impact of TBI [4]. For example, the incidence of TBI in countries such as Sweden, Italy, France, and Norway appears to have decreased over time, while the incidence of TBI in Spain and Taiwan may have increased [5,6]. Lower socioeconomic status, alcohol and drug use, and underlying psychiatric and cognitive disorders are also risk factors for head injury.

## 2. Diversity and Homologies in Adult Brains of Zebrafish and Mammals

While the embryonic vertebrate brain appears similar across species during early development, one can observe a surprising variability in adult brains regarding their morphology. Depending on the species, structures are different in size, cell type or density or some novel structures, while no obvious equivalent in other vertebrate species can emerge. Indeed, like the rest of the organs, the brain is shaped by evolution. For example, defining features of the mammalian thalamus proper, such as the thalamocortical connections, are inexistent in teleosts, as in the zebrafish model [7]. In fact, teleosts lack a sophisticated six-layered isocortex, which in mammals provides chief instances of sensory processing [8]. However, teleosts possess a dorsal pallial division that topologically corresponds to the mammalian isocortex [9]. Yet, the zebrafish dorsal pallium does not hold sensory areas that receive projections from relay stations comparable to the mammalian thalamus proper [10,11]. The auditory thalamic nucleus (CPo) of teleosts projects to the amygdala (Dm) and to the hippocampal (Dl) division, regions that are, like their mammalian counterparts, involved in emotional response behaviors and spatial orientation, respectively [12,13]. The thalamus is also less prominent in zebrafish than in mammals [14,15]. This is because teleosts have a preglomerular complex, an agglomeration of nuclei related to the posterior tuberculum, with its specific large dopaminergic cells [16], the paraventricular organ, and the posterior tuberal nucleus, which are absent in mammals [17,18]. The preglomerular complex is used as a sensory relay station in the zebrafish diencephalon [19].

A recent study identified the expression pattern of conserved genes in the pallium of adult zebrafish in order to define pallial subdivisions and to determine their homologs in tetrapods [20,21]. They suggested that the Dm corresponds to the pallial amygdala in mammals, the Dc is the homolog of the cortex and the Dl (ventral and dorsal parts) could be the homolog of the hippocampus [22]. Further evidence, by performing in situ hybridization for 1202 transcription regulators [23] (TRs) in the telencephalon of adult zebrafish, have identified particular regions displaying specific clusters of TR gene expression revealing similarities with the mammalian brain [24]. Among the 1202 TRs analyzed, 562 exhibited 13 distinct patterns, and some of them were restricted to the neurogenic niches localized along the telencephalic ventricle. In the same line, neurotransmitters, their synthesizing enzymes, and specific markers for GABAergic neurons in the telencephalon or serotoninergic innervation of the telencephalon (or serotoninergic neurons in other brain regions than the telencephalon) are expressed in a very similar way to what is observed in rodents, suggesting once again homologies with the mammalian telencephalic regions during development and adulthood [25,26,27,28,29,30]. In addition, concerning the visual pathways, it is generally admitted that jawed vertebrates generally possess two visual pathways to the pallium: a thalamo fugal-like and a tecto fugal-like. However, there is some variation in organization within Actinopterygii. Indeed, some species seem to have only one visual pathway to the pallium [31]. It has been described in cyprinids (goldfish and carp), which are, thus, relatively close to zebrafish. In holocentrids and gobies, which are related to Percomorpha and are distant phylogenetically from cyprinids, visual information is relayed by a different set of nuclei in the nucleus pre-thalamicus (PTh), which has not been identified in cyprinids. As its name indicates, it is located in a more rostral and dorsal position than PG, and it receives afferents from the optic tectum; injections in different parts of the pallium leads to the labeling of different parts of the PTh, as seen for PG [32].

## 3. Divergent Adult Neurogenesis in Zebrafish and Mammals

The number of newly produced neurons and neurogenic niches in the adult brain decreases during evolution; additionally, the regenerative potentiality dramatically declines. In mammals, adult neurogenesis is limited; two neurogenic regions are well described: the subventricular zone of the lateral ventricle (SVZ) and the sub-granular zone of the dentate gyrus in the hippocampus [33,34]. Moreover, newly formed neurons do not survive for a long time, likely due to a non-suitable local environment. In aged mice, the structure of the SVZ, the area of the adult CNS where the specific pool of neural stem progenitor cells (NSPCs ) is retained, is also profoundly altered [35]. The SVZ, ependymal and astrocytic cell morphology and functions undergo significant changes [36,37], and the RMS tends to disappear, as a consequence of the progressive decline in the number of neurons produced [38,39]. These changes correlate with the impairment of olfactory discrimination abilities [40]. Several in vivo studies have shown that the pool of actively proliferating NSPCs becomes dramatically reduced during aging, indicating that the curtailed neurogenic output of the aged SVZ is linked to the depletion of the NSPC reservoir. The decrease in the amount of NSPCs in vivo is paralleled by a weaker ability of the aged SVZ to form neurospheres upon dissociation and in vitro culture compared with the young adult SVZ [41]. This difference in neurosphere-forming ability was not detectable when the assay was performed with nestin-expressing cells, sorted from young adult and aged nestin:GFP mice, indicating that spared NSPCs from aged mice retain the competence to respond to growth factors in vitro [42,43]. Further in vitro experiments, however, support the hypothesis that the proliferative properties of NSPC, besides their numbers, change with the age of the mouse from which NSPC cultures originate. For example, a decline in bromodeoxyuridine (BrdU) incorporation has been observed in neurosphere cultures of young adult (2 months old) NSPCs compared with newborn animals [44]. A further decrease was found at 15 months. At this age, NSPCs also had lower survival rates and generated fewer neurons and more astrocytes, evidencing a generalized impairment of NSPC function at middle age [45,46]. Very recently, two studies reported opposite results in humans: by the Alvarez Buylla group, they concluded that neurogenesis drops to undetectable amounts during childhood [47,48], whereas a recent study reported lifelong neurogenesis [49,50]. Different from mouse and human, the brain of adult zebrafish exhibits a high number of proliferative areas mainly localized in 16 neurogenic niches [51] (Figure 1A,B). The majority are distributed along the ventricles of the telencephalon, diencephalon, and mesencephalon, from which newly differentiated neurons start to migrate [51,52,53,54,55]. The proliferative activity of stem cells in the cerebellum of zebrafish can be modulated by different growth factors. For example, the inhibition of fibroblast growth factor (FGF) signaling in vivo results in a marked reduction of stem cell activity [56]. In vitro studies using stem cells isolated from the proliferation zones in the dorsal telencephalon and cerebellum of a teleost fish confirmed the involvement of FGF in the control of mitotic activity [57]. A further aspect related to the regulation of proliferative activity concerns the period of transition of the adult stem cells from a quiescent state to an active state. During the quiescent state, the stem cells remain in the G0 phase of the cell cycle. After stimulation, they enter the G1 phase to continue generating new cells. In ventricular zones of the zebrafish telencephalon, where radial glial cells act as stem cells [58,59], the transition between these two states is regulated by Notch [60,61,62,63]. Notch induction can drive the stem cells into quiescence, whereas blocking of Notch re-starts cell division. Notch activation seems to arise from neighboring cells, which are potentially active progenitor cells.

Further studies concerning the mechanisms controlling adult stem cell activity in the CNS of teleost fish have been made by using the retina as a model system. In this study, one important aspect concerns possible differences in the genetic programs utilized during embryonic development versus post-embryonic development. A genetic screen in zebrafish has suggested that such differences exist [64]. All mutants isolated as part of this screen, the eyes presented normal embryonic development, but failed to add new cells to the post-embryonic neural retina from the CMZ. In detail, in this study, they isolated 18 mutant strains (1% of the 1740 F2 families screened) that exhibited normal embryonic development, but failed to add new cells to the post-embryonic neural retina. Other 46 lines (2.5%) with CMZ phenotypes were observed, but discarded for lack of specificity. They also observed six mutants with a subtly reduced CMZ. When they labeled these mutants with BrdU staining, they detected a significant reduction in cell division in the CMZ. This study also unearthed an important link, namely, that two neighboring proliferative populations, the CMZ and the peripheral RPE, may arise from a common stem cell in the larval eye. These evidence has also been reported by other studies in Xenopus and fish retina regeneration [65,66].

## 4. Neurotrophins during Vertebrate Evolution

Neurotrophin factors regulate neuronal differentiation and play key roles in neuronal survival, growth and plasticity. Since the discovery of nerve growth factor (NGF) in the 1950s [67], the family has progressively grown. Presumably, from two intermediate neurotrophin gene ancestors, the couple, NGF and NT3, as well as BDNF and NT4/5, were formed after the split of jawless fish, but before the split of cartilaginous fish from the common vertebrate lineage. In bony fish, an additional duplication was suggested: the same ancestor was the origin of duplication to NGF and NT 6/7, the latter a neurotrophin not known in other vertebrates [68]. Homologous genes that are derived by means of speciation are called orthologues, and if they are formed by gene duplication, they are called paralogues. NGF, BDNF, NT-3, and NT-4/5 are paralogues [69,70,71]. NT-4 was first isolated from Xenopus laevis [72] and NT-5 was first isolated from rat. Previous phylogenetic studies clearly show that NT-4 and NT-5 are orthologues. NT-6 and NT-7 have been suggested to be paralogues [70]. They probably resulted from the duplication of an ancestral fish NGF gene. Zebrafish NGF/NT-7, as well as trkB1/trkB2 and trkC1/trkC2 [73], were probably formed at the major duplication event early in the history of *Osteichtyes* [74].

In detail expression of trk receptors in cell types appears to be largely conserved among species. Photoreceptors express trkB and possibly trkC, bipolar cells express trkB and trkC, and retinal ganglion cells express trkB or trkCand, and to a lesser extent, trkA [75,76,77]. Many amacrine cells express trkB and, during development, trkA, and many horizontal cells express primarily trkA, and to a lesser extent trkB. Except for horizontal cells (trkA), the predominant trk receptor in the retina appears to be trkB. The patterns appear largely preserved among species. The fact that trkA expression can be increased in retinal ganglion cells after injury or during regeneration [76] might indicate that there is a latent potential for trkA expression that is normally suppressed. It is not known how the ‘normal’ retinal ganglion cells might differ from those that express trkA under certain conditions. Although trkB associates somewhat specifically with dopaminergic amacrine cells in some species, it seems to be expressed among a wider range of amacrine cell types in other species [78]. In addition to expressing trkB, a major fraction of developing amacrine cells also express trkA. The patterns of expression of trkA on the one hand and trkB/C on the other show interesting correlations between mammals and fish: trkA is primarily expressed in neurons with relatively large, overlapping dendritic fields and functions that have been categorized as ‘integrative’ [79], as opposed to neurons with relatively restricted dendritic and axonal fields and ‘analytical’ functions that correlate more with trkB and trkC expression. Indeed, in the cerebellum, the most ‘analytical’ and least ‘integrative’ part of the brain, there is no trkA expression, whereas those neuronal populations with overlapping inputs and more ‘diffuse’ and integrative functions (cholinergic basal forebrain, locus coeruleus, raphe, limbic, reticular neurons, nociceptive and sympathetic ganglia) are indeed the ones that express trkA [80].

## 5. Multiple Neurotrophin-Receptor Interactions in the Brain of Mammals after TBI

All neurotrophins are initially synthesized as inactive precursors or pro-neurotrophins, which are then processed by proteolysis to form the mature proteins [81]. The mature neurotrophin proteins form stable non-covalent dimers and initiate their biological actions by binding to one of two different classes of receptors: a high-affinity transmembrane tyrosine kinase receptor, Trk, and p75 neurotrophin receptor [82], which is a member of the tumor necrosis factor (TNF) receptor superfamily [83,84,85]. Three members of the TRK receptor family are known in mammals: TRKA, which binds NGF; TRKB, which binds BDNF and NT4; and TRKC, which binds NT3 [86,87] as represented in Figure 2. After that, a mature neurotrophin has been linked to its respective Trk receptor, the Trk receptor dimerizes and undergoes transphosphorylation to specific intracellular tyrosine kinase residues. These phosphorylated tyrosine-kinase residues act as docking sites for adaptor proteins that allow additional kinases to be recruited for activation of intracellular signaling pathways.

Activation of the Trk receptor is known to phosphorylate and activate PLC-γ1 and its associated second messenger signaling pathway [88]. PLCγ activity produces two different second messenger systems: IP3 and DAG, which increase intracellular release of calcium (Ca^2+^) and stimulate PKC-signaling, resulting in enhanced neuronal and synaptic plasticity. Trk receptors can activate Ras, a small GTP-binding protein that activates the MAPK signaling pathway to produce ERK. This phosphorylates and activates CREB, causing downstream transcriptional changes to genes that regulate neuronal differentiation and neurite outgrowth [89]. Another major pathway activated by Trk receptors is the PI3-K pathway, which leads to downstream activation of the serine/theronine kinase, AKT, to promote NF-kB-mediated neuronal survival [90]. The PI3K/AKT signaling pathway plays an important role to modulate cell growth, proliferation, and survival under physiological and pathological conditions [89]. In addition, PI3K/AKT signaling is also involved in axonal sprouting in cultured hippocampal neurons, which is an important mechanism underlying post-stroke functional recovery. In a previous study [91], the authors combined systemic administration of a high impact AMPAKine, CX1837, and local hydrogel delivery of BDNF, resulting in a synergistic increase in phosphorylation of AKT, MEK, and CREB in aged, two-year-old mice. This upregulation was observed in parallel to increased functional recovery of motor function. Further studies showed that modulation of GSK-3 after stroke can enhance axonal outgrowth [92]. Activation of the AKT/GSK-3/CREB pathway has also been observed in other age-related neurological disorders such as Alzheimer’s disease, with activation of this pathway leading to an improvement in cognitive function [93]. To this end, the full complement of Trk-mediated signaling pathways that play a role in post-stroke recovery, or their involvement in mechanisms associated with recovery, remain to be full elucidated.

While neurotrophins signal via two different receptor pathways, most of the attention has been focused on the activation of Trk receptors and their subsequent biological effects. However, over the past couple of decades, research has also begun to investigate the surprisingly diverse functions of p75NTR, including a somewhat paradoxical involvement in pro-apoptotic signaling [93]. Most neuronal populations that respond to neurotrophins co-express both p75NTR and Trk receptors. The exact mechanism remains elusive, the interaction between these two receptors is known to positively regulate Trk function following binding of a mature neurotrophin [91]. In the absence of Trk, p75NTR can also induce apoptotic signaling when a complex with sortilin is formed [94]; specifically, a Vps10-domain containing protein can act as a co-receptor for pro-neurotrophin binding. Next, the activated pro-neurotrophin-receptor complex activates Rac, a GTP-binding protein that induce the JNK cascade, leading to activation of pro-apoptotic genes and apoptotic cell death. Several previous studies in rodents showed an increase of gene and proteins expression of BDNF and TrkB following experimental TBI made by means of lateral fluid percussion [87], lateral gas pressure, or penetrating TBI [95]. In general, the impact of exogenous BDNF on neurogenesis was studied by infusion into the lateral ventricles of the adult rat. BDNF substantially increased the number of newborn cells in many regions, the preponderance of which differentiate into neurons. BDNF administration in the hippocampus was associated with an increased neurogenesis of granule cells in the dentate gyrus [96]. Further studies obtained different results, suggesting that BDNF, delivered intracerebroventricularly in mice and rats, failed to enhance neurogenesis in the subventricular zone, but even reduced it. These contradictory results could be due to differences in the reagents being used by the various labs [97]. It is known that the production of BDNF in some recombinant systems leads to misfolding of the protein and possibly changes in its efficacy to stimulate target receptors. In addition, differences in the abundance of the preprocessed pro form of BDNF relative to the mature form may also contribute to the differences in observed results. The results regarding the action of endogenous BDNF on neurogenesis are quite complex and contrasting [98]. Studies in heterozygous BDNF knockout mice reported that proliferation of neural stem cells was decreased in the dentate gyrus of the hippocampus. In contrast, conditional knockout mice with a depletion of BDNF in mature neurons exhibited an increase in hippocampal proliferation [99,100]. However, as a result of different experimental approach and time window of 1–72 h or 1 day–8 weeks from injury, BDNF-TrkB increased bilaterally and/or ipsilaterally quickly after TBI. This has also been confirmed in human, namely, in Vietnam combat veterans with focal penetrating TBI [95,101,102,103]. This mean that the BDNF-TrkB pathway is involved during the first response after an event of TBI. Further studies conducted in humans also observed conflicting results. For example, a cohort of children with severe head injury presented high levels of BDNF in cerebrospinal fluid (CSF) up to 24 h [104,105], which confirms previous data. Similar results were observed in a study in which patients presented higher levels of BDNF post-TBI versus controls, which was associated with a shorter time until death. On the contrary, one study showed that adult TBI patients had lower serum BDNF levels than healthy controls, and that lower BDNF values were also associated with incomplete recovery after TBI [106]. Moreover, it has been observed that NT4 can bind TrkB, as mentioned before, of which the values are increased after 1 to 23 h post TBI in human and mouse models [107]. Differently from these neurotrophins, previous studies have shown that NGF mRNAs and proteins are significantly elevated in the brain at 7 days post-TBI, and its specific receptor Trk-A is also expressed in non-injury sites, indicating that NGF responds to neurons in non-injured sites [108]. Further analysis reported that NGF increased in the cerebral cortex after brain trauma, and the activity of antioxidant enzymes also increased significantly [109]. Therefore, it is believed that NGF can play a role by inducing the synthesis of oxygen radical scavengers. For the influence of central cholinergic on the nervous system, it is suggested that NGF can reduce the demyelinating degeneration and necrosis of cholinergic neurons, promote the release of acetylcholine from hippocampal neurons, and improve the memory disorder formed after brain injury. Finally, further studies showed a significant decrease of NT-3 and TrkC receptor in mouse at 3 or 6 h post-TBI [110]. Other important factor not discussed here is the age of a patient or animal model. Age has been shown to influence the basal levels of some neurotrophins, such as BDNF, and may also influence the regulation and expression patterns of some neurotrophins after brain injury [111,112]. Another less-discussed topic is the differential expression levels and functions of mature and pro-neurotrophins. For example, pro-NGF and pro-BDNF have been shown to increase soon after pilocarpine-induced seizures in the hippocampus in vivo. This increase was associated with neurons and reactive astrocytes, but not with microglia [113].

## 6. Neurotrophins and Their Receptors Expression after TBI in Zebrafish Models

Recent studies described the expression pattern of different neurotrophin and receptors in larva and adult zebrafish brain in physiological condition and after TBI. In detail, in adult zebrafish, BDNF expression had a similar distribution than in the larvae with the most prominent staining in dorsal telencephalon, area preoptica, dorsal thalamus, posterior tuberculum, hypothalamus, synencephalon and optic tectum. A more diffuse and weaker labeling was detected in other brain regions. This study also characterized the phenotype of cells expressing BDNF mRNA by means of double staining. Interestingly, they could not detect BDNF/Aromatase B or BDNF/BLBP co-expression, suggesting that radial glial cells do not express BDNF under physiological [114] conditions. Similarly, double staining with the cell proliferation marker PCNA failed to show any co-expression of BDNF and PCNA in the same cells [115]. In contrast, double staining with neuronal makers, notably Huc/d, MAP2, and acetylated-tubulin, clearly identified BDNF-expressing cells as neuronal cells in the brain of adult zebrafish [116]. Further studies reported the distribution of TrkB receptor in different regions of zebrafish brain [117]. After TBI on adult zebrafish, a significant increase of BDNF mRNAs levels in homogenates of the whole lesioned telencephalon occurred at 1 day post lesion (dpl), compared to control unlesioned animals. Then, BDNF mRNA levels decreased with time after lesion. BDNF mRNA-expressing cells consisted of neurons, and their number increased mostly immediately after injury. The receptor TrkB increased at 24–48 h after injury, and this increase was sustained during the following 14 days [116]. Data by RNA-seq analysis approach confirmed the expression of other neurotrophins and their receptors after TBI in adult [118] zebrafish. In detail for BDNF, NT4 increased quickly, similar to the receptor TrkB after TBI. Recent studies also described the expression pattern of NGF and its receptor TrkA in adult zebrafish brain under physiological condition and after TBI [119,120]. These studies evidenced that NGF is specifically expressed in mature neurons, and an interesting point is that NGF and its receptor TrkA increased after 7–10 days after TBI. This means that the NGF-TrkA pathway could be involved in a time when the repair is completed. Finally, NT3 and its receptor TrkC have been identified in adult zebrafish brain, and a recent study reported that it can be involved in spinal cord repair after injury [121]. Further investigations concerning this pathway after TBI in adult zebrafish could clarify its role. Figure 3 shows, at all time points, the expression of each neurotrophin and specific receptor after an event of traumatic brain injury in human, mouse, and zebrafish.

## 7. The Use of Neurotrophins in Regenerative Medicine: A Promising Therapy 

Neurodegenerative diseases, such as injury, remain the cause of adult death and disability worldwide. With the aging population on the rise, and these diseases preferentially affecting the elderly, there is a growing interest in the development of interventions that can fight against neurological decline and enhance functional recovery. Therapies for brain injury are limited due to the narrow window of opportunity that is required for success. Therefore, the practice of regenerative interventions have become of highest priority to help treat brain injury at a clinically relevant time point, and to successfully translate findings from bench to bedside. After decades of preclinical studies and still no successful treatments for injury recovery in humans, researchers have agreed that future work investigating the neurobiology and progression of post-injury recovery (both spontaneous and treatment-induced) will be crucial for the journey toward the identification of key molecular targets and an optimal time of intervention [122]. For example, it has been observed that molecular inhibition of pro-NGF binding to p75 after experimental TBI has been shown to improve the outcome in mouse TBI models [123]. In a mouse model of heart ischemia-reperfusion injury, expression of pro-NGF increases in cardiomyocytes, while expression of p75 increases in microvascular pericytes, which results in microvascular damage. Therefore, it becomes apparent that, during the course of TBI [124], the effects of signaling through p75 are relevant not only to neural cells, but also to vascular cells, as binding of pro-neurotrophins produced during this process may undermine vascular supply and the generation of activated pericytes that act as medicinal signaling cells. During this process, endothelial cells are also expected to be compromised as they lose pericyte support [125]. In line with this, it has been found that increased levels of the endothelial cell markers factor and matrix metalloprotinase-9 in the blood after TBI are inversely correlated with the outcome. This means that a possible interaction of different cells of the blood–brain barrier can modulate neurotrophin-receptor pathways.

Again, different studies in preclinical animal models of neurodegenerative disease showed that neurotrophins can modulate the functional repair after brain injury. However, exogenous administration of these neurotrophins is limited by a lack of blood–brain barrier (BBB) permeability and a rapid degradation. Indeed, different clinical trial studies published in the past, also involving stem cell therapy, have tried to infiltrate the BBB by intracerebroventricular (ICV) injection of neurotrophins, but they evidenced several problems [126]. The use of innovative approaches such as biomaterials could offer a mechanism to penetrate the BBB and treat the injured brain with concentrations that are only a fraction of those used during systemic administration [125,127,128]. In detail, these studies have shown great promise using biomaterial delivery systems to deliver neurotrophins to enhance functional recovery, with each biomaterial type yielding distinct properties that suit different biomedical applications. However, given the complex spatiotemporal, pathological events that occur in the time following neurodegenerative diseases, these delivery systems will yield great promise with the delivery of multiple neurotrophins to appropriate regions of the brain at relevant therapeutic time points to maximize recovery [93,129].

## 8. Concluding Remarks

The evidence in this present review suggests that each neurotrophin-receptor pathway can play a specific role after traumatic brain injury in both zebrafish and mammals. Given the knowledge we have already gained around the neurotrophic pathway, especially with respect to BDNF-mediated improvements in functional recovery and the innovative discovery made in the biomaterial-delivery field, there is an exciting prospect of being able to deliver a potential treatment to contribute toward recovery from injury and other neurological disorders. To ensure such treatment options translate into the clinic, much needed follow-up studies are required and clinical validation studies are to be undertaken. It is highly known that regenerative processes after damage are extremely scarce in the mammalian brain; in contrast, the zebrafish brain exhibits a high number of proliferative areas that permit to regenerate after TBI. For this reason, further studies using a zebrafish model could open new therapies to promote brain repair not only after an event of TBI, but also after brain lesion induced by neurodegenerative diseases.

## 9. Materials and Methods

### 9.1. Literature Search

Comprehensive literature was searched in three different databases: Pubmed, Web of Science, and CENTRAL, which was were performed in December 2020. The filters employed in the searched databases were language (English) and date of publication (1960–2020). Search terms revolved around variations of the following terms: “neurotrophins,” “traumatic brain injury,” “clinical therapy for TBI,” “neurogenic niches,” “zebrafish brain,” and “mouse brain.” I focused on articles that concerned the role of neurotrophins during an event of TBI. All citations have been imported by using EndNoteX9 (license University of Geneva).

### 9.2. Figures 

All figures have been generated by using Adobe Photoshop.

## Figures and Tables

**Figure 1 ijms-22-01585-f001:**
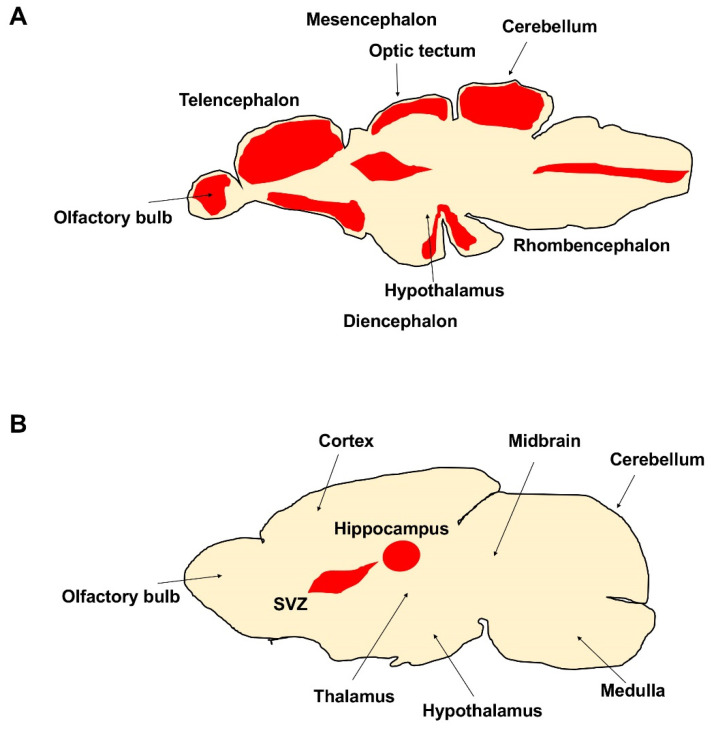
Different neurogenic niches (in red) in the brains of adult (**A**) zebrafish and (**B**) mammal, SVZ is the subventricular zone of the lateral ventricle.

**Figure 2 ijms-22-01585-f002:**
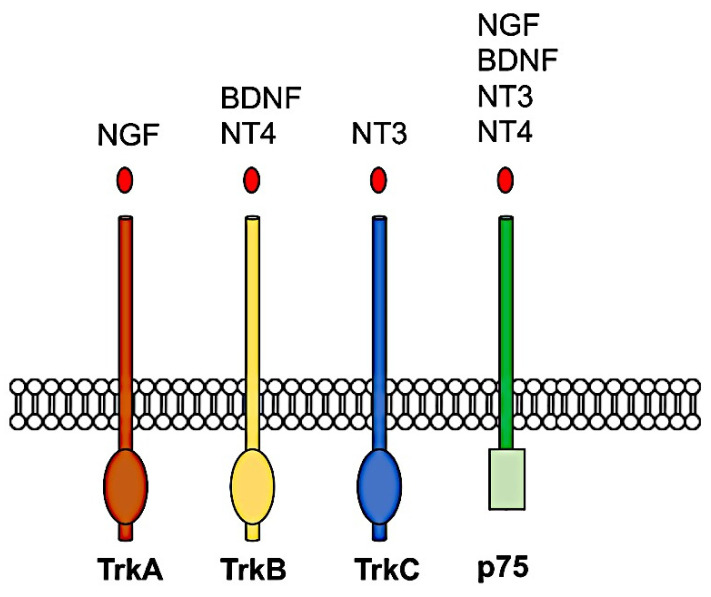
Neurotrophin receptor pathways.

**Figure 3 ijms-22-01585-f003:**
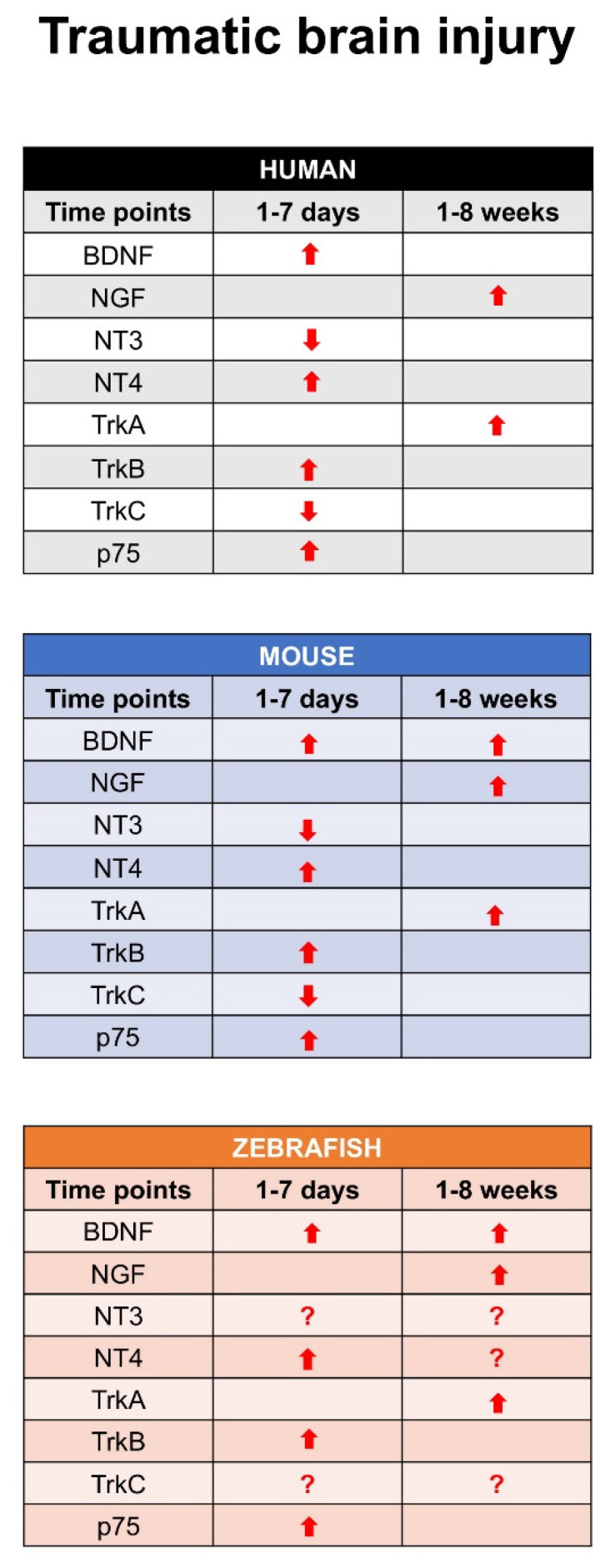
Time point expression of all neurotrophins and their receptors after traumatic brain injury in human, mouse, and zebrafish. All arrows (⇧) indicate that the mRNA or protein expression increase, or decrease (⇩). Question marks (?) mean that literature are not present studies, and could be interesting to investigate.

## Data Availability

Not applicable.

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
