# Peer review of "Neurotrophins Time Point Intervention after Traumatic Brain Injury: From Zebrafish to Human"

_ijms, 2021, doi:10.3390/ijms22041585_

Round 1

Reviewer 1 Report

Cacialli summarizes in this review, the current knowledge and future perspectives of the specific role of each Neurotrophins after TBI. The author described the differences and the homologies between adult zebrafish and mammals' brains, and furthermore, he reported the specific role and time point of intervention for each neurotrophic factor and receptor in zebrafish and mammals after an event of TBI. The article gives an interesting widespread and scientific perspective on a field that has been developed.  The work of Cacialli is important and exciting for a broad community, therefore, deserves publication after some modifications.

Comments:

  1.   From line 114 to line 120: I would suggest expanding what was found in the work of Wheman et al (ref.58).
  2.  It is recommended to Add a Material and Methods section in order to describe how the literature search was performed.
  3.   It is suggested to add a schematic figure about the signal transduction pathways described in the paper.

Author Response

Reviewer 1

Cacialli summarizes in this review, the current knowledge and future perspectives of the specific role of each Neurotrophins after TBI. The author described the differences and the homologies between adult zebrafish and mammals brains, and furthermore, he reported the specific role and time point of intervention for each neurotrophic factor and receptor in zebrafish and mammals after an event of TBI. The article gives an interesting widespread and scientific perspective on a field that has been developed.  The work of Cacialli is important and exciting for a broad community, therefore, deserves publication after some modifications.

Thank you for the positive evaluation of my work.

1.  From line 114 to line 120: I would suggest expanding what was found in the work of Wheman et al (ref.58).

Thank you for this suggestion. I improved in the new version, a detailed description of this important study. 

2. It is recommended to Add a Material and Methods section in order to describe how the literature search was performed.

In the new version of the manuscript, I added a specific section of Material and methods, as suggested by this reviewer.

3.  It is suggested to add a schematic figure about the signal transduction pathways described in the paper.

In the new version of the manuscript, I added a schematic figure (Figure 2) of neurotrophin-receptor pathway described.

Reviewer 2 Report

In this review paper, the author describes in a thorough and elegant manner, a parallel between zebrafish and mammals with respect to regenerative properties after traumatic brain injury, with a particular focus on neurotrophins. The topic is of interest to IJMS readers. The manuscript is well written and easily readable. The structure is generally correct. References are accurate and findings are discussed with balance.

Minor point:

  • While the abstract starts pointing out the epidemiological relevance of TBI and the very purpose of this review, there is not a corresponding section in the main text. In my view, the manuscript would gain in interest if a brief description on the impact of TBI would be added at the beginning

Author Response

Reviewer 2

In this review paper, the author describes in a thorough and elegant manner, a parallel between zebrafish and mammals with respect to regenerative properties after traumatic brain injury, with a particular focus on neurotrophins. The topic is of interest to IJMS readers. The manuscript is well written and easily readable. The structure is generally correct. References are accurate and findings are discussed with balance.

Thank you for the positive evaluation of my work.

Minor point:

  • While the abstract starts pointing out the epidemiological relevance of TBI and the very purpose of this review, there is not a corresponding section in the main text. In my view, the manuscript would gain in interest if a brief description on the impact of TBI would be added at the beginning

Thank you for this suggestion, in the new version of the manuscript, I added a brief description of TBI incidence in the world.